# Microfluidic control over topological states in channel-confined nematic flows

Simon Čopar [1], Žiga Kos [1], Tadej Emeršič[2,3] & Uroš Tkalec [2,3,4]*

Compared to isotropic liquids, orientational order of nematic liquid crystals makes their rheological properties more involved, and thus requires fine control of the flow parameters to govern the orientational patterns. In microfluidic channels with perpendicular surface alignment, nematics discontinuously transition from perpendicular structure at low flow rates to flow-aligned structure at high flow rates. Here we show how precise tuning of the driving pressure can be used to stabilize and manipulate a previously unresearched topologically protected chiral intermediate state which arises before the homeotropic to flow-aligned transition. We characterize the mechanisms underlying the transition and construct a phenomenological model to describe the critical behaviour and the phase diagram of the observed chiral flow state, and evaluate the effect of a forced symmetry breaking by introduction of a chiral dopant. Finally, we induce transitions on demand through channel geometry, application of laser tweezers, and careful control of the flow rate.

[1] Faculty of Mathematics and Physics, University of Ljubljana, Jadranska 19, 1000 Ljubljana, Slovenia. [2] Faculty of Medicine, Institute of Biophysics, University of Ljubljana, Vrazov trg 2, 1000 Ljubljana, Slovenia. [3] Jožef Stefan Institute, Jamova 39, 1000 Ljubljana, Slovenia. [4] Faculty of Natural Sciences and Mathematics, University of Maribor, Koroška 160, 2000 Maribor, Slovenia. *email: uros.tkalec@mf.uni-lj.si

                                   1

Many advances in laboratory science move toward automatisation, miniaturisation and replacing requirements for expensive immobile equipment with integrated microfluidic platforms. Handling of liquids on micrometre scale poses challenges in mixing, pumping and directing, which have all been a focus of intense research in the past two decades[1–3]. Microfluidic environment also restricts the fluid to a very particular and well-defined rheological regime, so instead of using the liquid as a transport medium in a device with a higher purpose, we can focus on systematically studying and observing properties of the fluid itself. This is particularly useful considering the growing body of research on non-Newtonian, anisotropic and active fluids[4–8]. Complex fluids exhibit a plethora of hydrodynamic regimes and transitions when subjected to shear flows, usually found in microfluidic environments, and possess many internal parameters and degrees of freedom to be observed or measured[9–12]. Compared to conventional isotropic fluids, complex fluids require much finer control in order to observe the desired non-equilibrium states in shear flow. In living cellular matter, and various types of recently studied active materials, internal driving forces induce collective motion, which can be described to a great extent as emergent continuous rheological regime, from spontaneous laminar shear flow[13–15] to irregular behaviour with persistent creation and annihilation of topological defects and creation of dynamical domains[16–19]. In passive liquids, nontrivial rheological properties can likewise be probed by finely tuning external fields and subjecting the liquid to different constraints by means of confinement in a microfluidic environment[20,21].

Nematic liquid crystals (NLCs)[22] are fluids with orientational order of their anisotropic building blocks, which affects their rheological behaviour when confined to thin planar cells or a network of channels. Confined nematic microflows are of use in optofluidic applications[23–25] and for guided material transport through microfluidic networks[26,27]. Coupling between nematic orientational order and the flow drives structural organisation and hydraulic conductivity in porous networks[28–30], and leads to shaping of the molecular field by Poiseuille flows[31–35]. Flow-induced deformation of the nematic texture was studied also in chiral nematics[36–38] and blue phases[39,40]. Interestingly, chiral order can emerge also in intrinsically achiral liquid crystals[41–43], which is typically due to exclusion of certain elastic deformation modes and of the confinement[44–47]. It is of particular interest if such chiral instabilities occur also out of equilibrium, i.e. due to interaction with flow, and how such chiral modes couple with previously observed structural transitions and textures in Poiseuille flows of nematics[48–56].

In this paper, we study a NLC flowing in thin microfluidic channels with perpendicular surface anchoring of nematic molecules on the channel walls. We employ precise regulation of the driving pressure, design of channel geometry, and laser tweezers to stabilise and reliably reproduce a previously unreported topologically protected flow regime with a broken mirror symmetry. Alongside experimental demonstrations, we shed light on the mechanisms that connect the behaviour of the NLC with its material parameters through use of numerical modelling. Based on geometric and topological assumptions, we construct a phenomenological Landau model that parametrises the flow profiles and faithfully reproduces the phase behaviour of the achiral-to-chiral transition. Using experiments, simulations and theory, we investigate the symmetry-breaking transitions, especially a transition into a hidden non-equilibrium chiral nematic state, and provide descriptions of their behaviour, stability and dependence on geometric and material parameters.

The paper is organised as follows: In the first section, we experimentally realise and describe all flow states in a rectangular channel at different flow rates. In the second section, we proceed with manipulation of the flow profiles via channel geometry, application of laser tweezers and symmetry breaking by addition of a chiral dopant. In the third section, we provide theoretical support for the description and classification of observed rheological regimes and reinforce them with numerical simulation of director profiles in relation to the flow rate and elastic constants. Finally, we present a minimal phenomenological Landau model that reproduces the simulated phase behaviour and discuss the general behaviour of flowing anisotropic liquids.

## Results

**Identification of flow regimes**. In the first experiment, we studied the flow of a NLC in a linear homeotropic channel with a rectangular cross section (see Methods). We gradually increased the flow rate to $\approx 0.5\ \mu L\ h^{-1}$ and observed the orientational phase transition where molecules turn from initially vertical homeotropic state to a flow-aligned state in which the director is aligned in the flow direction (Fig. 1a–e). In the weak flow, the uniform homeotropic ($H$) director (Fig. 1a) becomes only slightly bowed towards the flow direction—here referred to as $B$ state (Fig. 1b), causing the appearance of birefringent colours under crossed polarisers (Fig. 1e). Under strong flow, the nematic undergoes a discontinuous transition into a flow-aligned state—here dubbed dowser ($D$) state (Fig. 1d), in analogy with other similar systems[33,57]. In fact, the flow-aligned nematic profile is analogous to the escaped radial nematic profile, typically observed in nematic-filled capillaries[21]. The states are topologically distinct, as in $B$ state the director makes no net turn between the glass plates, but in $D$ state it makes a half-turn. The interface between the states is traced by a disclination line, as seen in Fig. 1e. The shape of the interface depends on the aspect ratio of the channel, and the boundary conditions at the side walls of the channel, e.g., the transition front may propagate from the sides instead of the center[58]. This observed sequence of states is generic and well understood as it has been widely studied in experiments[31,34,52] and numerical simulations[54–56]. In even stronger flow, driven by shear test apparatus or syringe pumps, oscillatory instabilities develop and the steady-state assumption is unjustified[49–51].

However, if we quickly and steadily increase the flow rate, we observe an instability that shows as colour variations under the microscope, before the occurrence of the flow-alignment transition. As we will elaborate on in detail later, the $B$ state undergoes a continuous symmetry breaking transformation into another, chiral ($B^*$) state with a left- and right-handed realisation ($B^+$ or $B^-$ for specific handedness), presented in Fig. 1f–h. The transition may nucleate from one side of the channel or randomly in the bulk from nematic fluctuations at certain flow velocity (Fig. 1f, g), and the fluctuations gradually grow into left- and/or right-handed mesoscopic domains of distinct birefringent colours (Supplementary Movie 1). The subsequent $B^*$ domains of opposite handedness refuse to merge, instead forming a soliton-like structure between them (Fig. 1g). The orientational phase transition may also start from the sides of the channel and propagate to the middle, where growing chiral nematic domains collide in a flexible longitudinal soliton (Fig. 1h and Supplementary Movie 2). In an achiral medium, the left and right-handed $B^*$ domains are degenerate, so the central longitudinal soliton is stable, but prone to long-wavelength undulations (Fig. 2a). The solitons in the $B^*$ phase themselves act as line defects with small but finite line tension, which is clearly demonstrated in the two-dimensional (2D) sessile drops running along the edges of the channel (Fig. 2b). This clearly demonstrates discrete symmetry breaking into a pair of topologically protected chiral states, which we will discuss in the following sections.

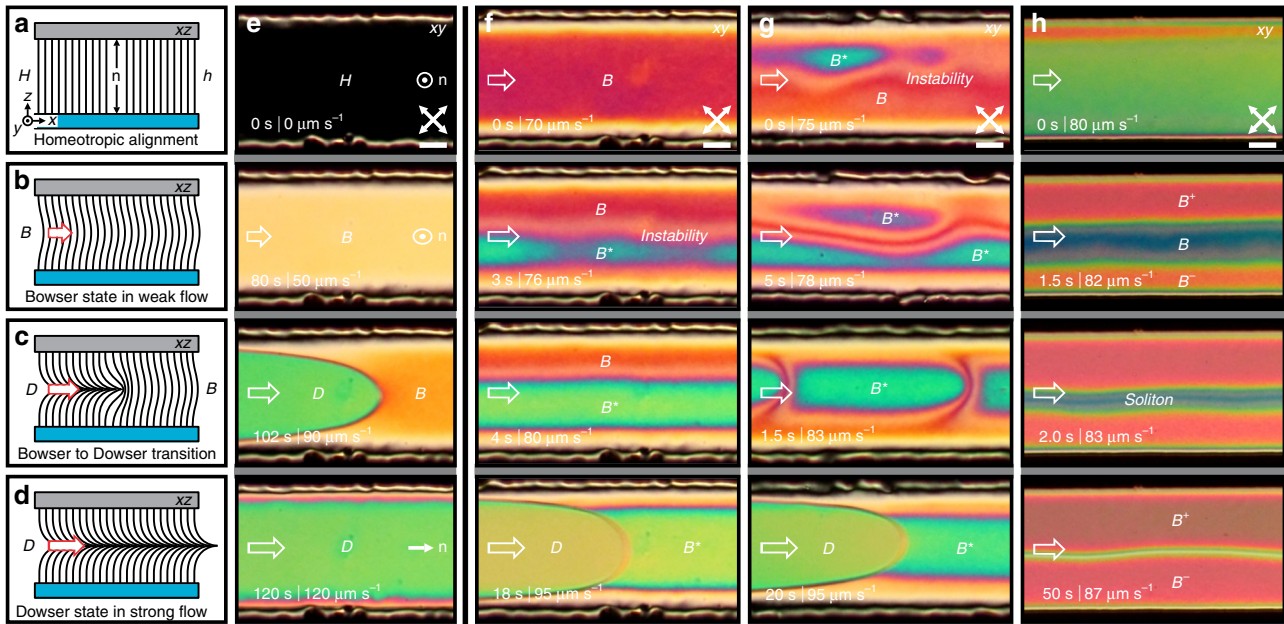

**Fig. 1 Orientational phase transitions in pressure-driven nematic flows. a** Side view of equilibrium homeotropic (*H*) state where the director **n** is vertically aligned between the top PDMS and bottom glass substrates; *h* designates the height of a microchannel. **b–d** The bowed homeotropic – *B* state – is stable under weak flow and undergoes discontinuous transition into the flow-aligned – *D* state – with an increasing flow velocity. **e** A sequence of experimental images shows top view of the *B* to *D* state transition. The flow acceleration is gradual and relatively slow ($\approx 1\ \mu m\ s^{-2}$). **f–h** Faster increase of flow velocity ($\approx 15\ \mu m\ s^{-2}$) makes the *B* state locally unstable as it breaks the symmetry into the chiral $B^{*}$ state of left- and/or right-handed domains (denoted by $B^{\pm}$ to highlight the opposite handedness and by $B^{*}$ where handedness cannot be determined from the micrographs). This emergent transition is continuous and fundamentally different from the transition to *D* state which involves a disclination line between differently oriented regions. A more uniform transition to $B^{*}$ state involves the growth of oppositely handed $B^{\pm}$ domains from the channel walls, creating a flexible soliton in the middle. More details of the transition dynamics are shown in Supplementary Figs. 1 and 2. White empty arrows on the left side of panels indicate direction and qualitative flow velocity throughout the paper; its approximate value is written in the corners of polarised micrographs. White double arrows show the orientation of the polarizers. Scale bars, 20 μm.

The observed chiral states are challenging to create, maintain and manipulate in microfluidic environment as they require fast, precise, and pulsation-free flow modulation that we repeatedly achieved by using piezoelectric regulated flow control system (see Methods). In general, it takes only few seconds to reorient the nematic medium from its initial *H* or *B* state to $B^{*}$ domains at flow velocities that are close to discontinuous transition into the flow-aligned *D* state (see Supplementary Note 1). Then, the flow rate has to be adjusted carefully to preserve the emergent chiral state for prolonged time, e.g. a couple of minutes. The sequence of state formation and stability depends on how quickly the flow rate changes, seen from stability thresholds during stationary flow (Supplementary Fig. 1) and state lifetimes when pressure is gradually accelerated over the wide operating range of the pressure controller (Supplementary Fig. 2). Sensitivity of the $B^{*}$ orientational phase to minute flow irregularities suggests that the $B^{*}$ state appears in the 'supercooled' regime where the bulk *D* state is already stable. The *B* to *D* transition is a first-order phase transition and requires an activation energy to trigger, so such hidden metastable regimes are possible. Conversely, appearance of a fluctuating grainy texture in the previously smooth *B* phase (Fig. 1g and Supplementary Movie 1) is a hallmark of a second-order phase transition, suggesting that the transition from *B* to $B^{*}$ is continuous. We further observe that tiny details, such as nonuniform surface anchoring, asymmetry in flow profile or slightly biased flow velocity, can additionally affect which type of continuous symmetry breaking will occur in a channel (Fig. 1f–h). We suspect that technical characteristics and significantly slower response time of syringe pumps that have been used in similar recent studies[31,34,58] may be one of the reasons why these

phenomena remained unexplored. Though Sengupta et al.[31] have observed and identified complex bend and splay deformations of the director in a medium flow regime inside homeotropic channels, they did not elaborate further on their dynamic formation.

Spontaneous emergence of chiral structures in achiral nematics in response to geometry—around inclusions, capillaries, and tori —has recently attracted attention of several groups[43–47]. These results clearly show the importance of elastic anisotropy, geometry and boundary conditions on twisted ground state configurations in a variety of confined materials and elucidate mechanisms of multiple symmetry breaking in a process of topological defect formation. To the best of our knowledge, orientational phase transitions with chiral instabilities in homogeneous nematic flows have not been discussed in the framework of experiments, theoretical modelling or numerical simulations, though localised twist deformations in an electric field have been recently demonstrated in different setups and contexts—in 2D Poiseuille flows under effect of an electric field, measured by Pieranski et al.[32], or by backflows caused by pulsed electric fields in 2D pattern formation by Migara et al.[59]. The recent publication by Agha and Bahr[60] reports wedge to twist to zigzag structural transformation of nematic disclination lines under the influence of anchoring, flow and electric field.

**Stabilisation of chiral flow states**. Instead of stabilising the $B^{*}$ state through control of the driving pressure, flow velocity can be modulated spatially by varying the channel shape. At high enough flow velocities, the $B^{\pm}$ states with a central soliton can break

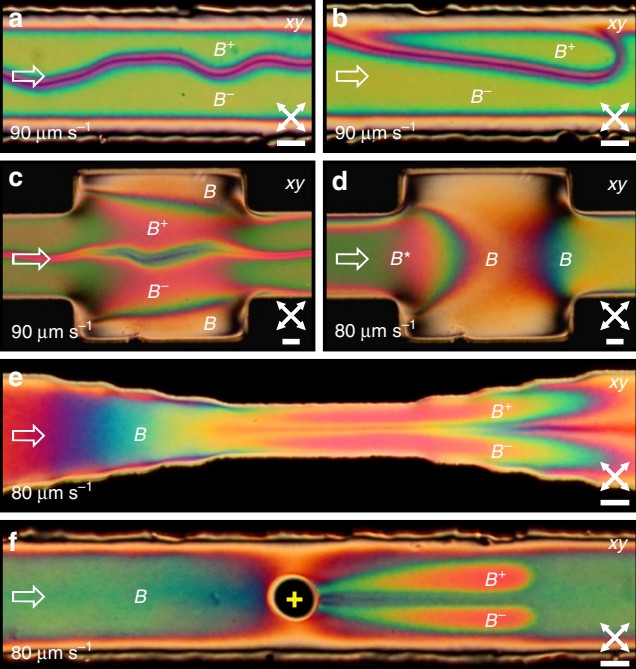

**Fig. 2 Manipulation and stabilisation of chiral $B^{\pm}$ states with channel geometry and laser tweezers. a** Flow-stabilised achiral soliton is oscillating in the middle of the channel. **b** A droplet of oppositely handed domain is running along the side of the channel. **c** A wider square chamber, which $B^{\pm}$ states have to breach to reach the other side. Note the increase of the soliton width inside the square chamber due to smaller flow magnitude. **d** At slightly lower flow velocity, the $B^*$ state is stopped by the channel widening. **e** A doubly tapered channel increases the flow velocity continuously, reaching the conditions for the emergence of $B^{\pm}$ states near the middle, which is then advected to the wider part of the channel. **f** Similar phase splitting can be achieved by a laser spot (yellow cross), which is locally heating the flowing nematic and consequently affecting the nematic ordering, triggering the $B$ to $B^{\pm}$ transition. Scale bars, 20 μm.

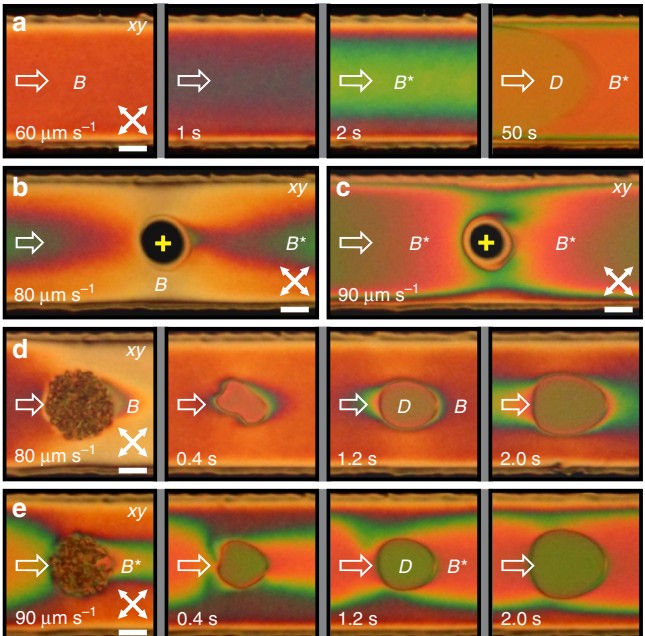

**Fig. 3 Formation and stability of chiral $B^*$ phase in flow of a weakly chiral nematic. a** Gradual transition from $B$ phase to chiral $B^*$ phase occurs at lower flow velocity and typically starts from the center of a channel. The small amount of right-handed chiral dopant triggers continuous growth of a single chiral domain with no soliton that is eventually transformed into flow-aligned $D$ state. Note that with introduction of intrinsic chirality, the distinction between $B$ and $B^*$ states becomes blurred. **b** A static laser beam in a flowing $B^*$ phase generates a small isotropic region with different rheological properties that allows faster flow, slowing down the flow in the surrounding medium, which thereby transitions into the $B$ state. **c** At higher flow velocity, the laser-induced isotropic island cannot split the wrapping $B^*$ phase as it becomes sufficiently stable over long time and length scales. **d** Laser-induced nucleation of $D$ domain in the $B$ phase. While the domain is growing, it is getting surrounded by greenish-coloured $B^*$ phase. **e** A similar experiment at higher flow velocity, where the flow is already in the $B^*$ state before the quench, creating another $D$ domain that is stabilised inside the $B^*$ phase. Scale bars, 20 μm.

through 200 μm long and wide expansion chamber (Fig. 2c), whereas at slightly lower flow velocity the $B^*$ state without a soliton is stopped and continuously transformed into the $B$ state inside the same widening of the channel (Fig. 2d). Thereby, such expansion chambers can be used to observe the persistence of the $B^*$ phase when injected into the region of $B$ phase at different flow velocities. Conversely, inside a continuously tapered channel, the $B^*$ phase with a central soliton can be induced in the narrowing (Fig. 2e). Furthermore, laser tweezers can be applied not only to nucleate $D$ domains, as shown in our previous work[35], but to manipulate the delicate balance between $B$ and $B^*$ states close to the phase transition. In Fig. 2f, one can observe how a flowing nematic in the $B$ phase immediately transforms into an opposite pair of $B^*$ states after traversing locally heated region (black spot), produced by a laser light absorption in the conductive substrate (see Methods). The applied temperature gradient affects the viscosity of the nematic fluid and thus the flow profile, and leads to a strong gradient in the nematic ordering. This change in conditions generates the $B^*$ state that persists for several hundreds of microns downstream and then gradually transitions back to the $B$ state, which is stable at the conditions of an undisturbed channel at the ambient temperature and chosen flow velocity. The demonstrated approach is useful to generate segments of chiral phase in a controlled way at will in any channel.

The symmetry between the $B^{\pm}$ states of the $B^*$ phase can be broken by introducing a chiral dopant into the system. To induce

the difference in free energy between the states of opposite handedness, we added 0.08 wt% of CB15 to the 5CB NLC (see Methods). In a weakly chiral sample, transition into the $B^*$ state becomes smoother and practically uniform, no sharp phase transition is observed and no solitons appear due to one chirality being favoured over the other (Fig. 3a and Supplementary Movie 3). In terms of phase transitions, the effect of a dopant is similar to the effect of external magnetic field on a ferromagnet. The transient $B^*$ region arises from the middle of a channel and gradually extends toward the walls. Its width can be tuned or maintained by adjusting the flow rate. By using laser tweezers, one can affect the flow profile and induce stronger chirality downstream and less chiral ($B$-like) regions in the transverse direction (Fig. 3b, c), similar to what we observed in achiral nematic flow at similar flow velocity (Fig. 2f). Finally, we also nucleated small $D$ domains in the $B$ and $B^*$ states of a chiral nematic flow by quenching an isotropic island to the nematic phase (Fig. 3d, e). The formation is similar to the formation in pure 5CB flow[35] as the $D$ state evolves with the flow and supplants the surrounding phase.

**Classification of flow-induced patterns**. States of director field in the microfluidic channels are composed of uniform regions

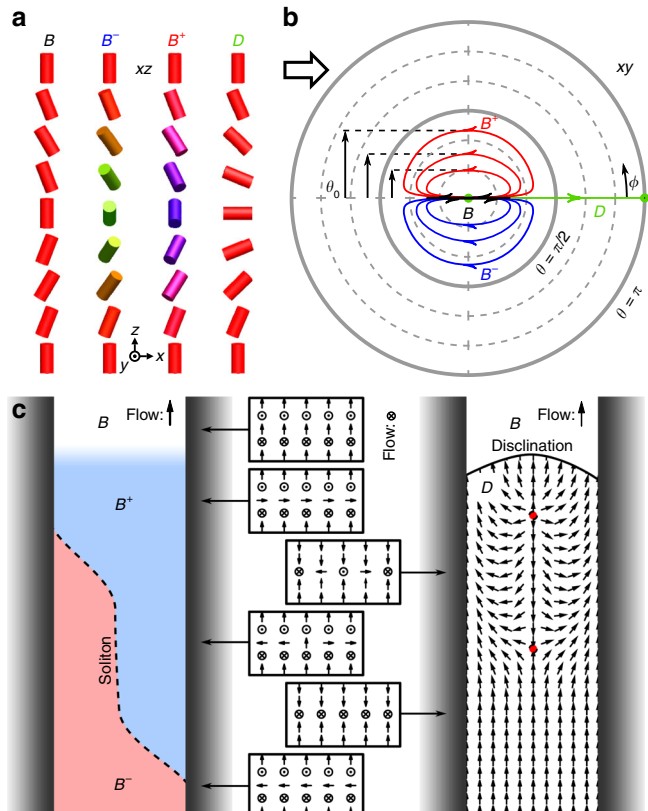

**Fig. 4 Idealised schematics of flow-induced director profiles. a** Typical columns of vertical cross section of each state, showing the characteristic director profile. $B^\pm$ states are mirror images of each other. **b** Polar plot, representing the directions traced by the four states of flow when $z$ coordinate traverses the height of the channel. $B$ state only deviates back and forth in the direction of flow, chiral $B^\pm$ states deviate sideways, tracing a clockwise or anticlockwise path when viewed from the top, and the $D$ state completes a full $\pi$ turn. The mid-cell deviation $\theta_0$ measures the degree of symmetry breaking in the chiral $B^*$ phase (denoted by black vertical arrows). The sketch depicts a few examples of how the profile behaves at different amplitudes and handedness. See Fig. 5 for quantitative simulation results. **c** Schematic depiction of transitions and solitons between the observed states. Left: the achiral $B$ state transitions into the chiral $B^\pm$ states via a second-order phase transition. Transitions between $B^\pm$ states are carried by nonsingular solitons (dashed line). Right: transitions from all $B$-type states into the $D$ state are first-order and are separated by a disclination line. The dowser state is a polar quasi-planar state, which contains its own dynamics of point defects (red dots).

in which horizontal variation of the director is mostly negligible, so for theoretical description, it is sufficient to only consider the director profile in a vertical direction. Each state is codified by the properties of a director in a vertical 'column', parametrised with polar angles as $\mathbf{n}(z) = (\cos\phi \sin\theta, \sin\phi \sin\theta, \cos\theta)$ in relation to the vertical coordinate $z$ between $\pm h/2$, where $h$ is the channel height. This column is subjected to the homeotropic boundary condition at the top and the bottom substrate (Fig. 4a).

Qualitatively the topological differences between the states are entirely encoded by how they reconcile the boundary condition $\theta(\pm h/2) = 0$. The absolute ground state for zero flow is the uniform $H$ state where $\theta(z) = 0$. With increasing flow, the director aligns more strongly along the channel due to shearing influence of the Poiseuille flow, resulting in the $B$ state with $\theta(z)$ deflecting back and forth, while the polar angle remains at $\phi = 0$

(Fig. 4a, b). However, the boundary conditions are also met if the director escapes out of the plane defined by the vertical and the flow direction, as it does in the $B^\pm$ states.

The escape can happen in two ways that are mirror images of each other, and are seen as clockwise and anti-clockwise path traced by the tip of the director when seen in the top projection (Fig. 4a, b). The magnitude of this symmetry breaking is best measured by the mid-plane deflection from the vertical, $\theta_0 = \theta(0)$, seen in the polar plot as the width of the ellipse-like path in Fig. 4b, with the sign distinguishing the chirality. In the $B^*$ state, the director in the mid-plane can be perpendicular to the flow direction in two ways, which cannot exist next to each other without an achiral $B$ region between them. As long as elasticity provides an energy barrier, the states are topologically protected and the transition is confined to a narrow achiral soliton.

The topological transition into the flow-aligned $D$ state is reached at high flow rates. In this state, the $\theta$ angle makes a $\pi$ rotation across the thickness, which is fundamentally incompatible with zero overall rotation in the $B$ and $B^*$ states, so a singular disclination line must exist at the boundary of this state. All the states are represented schematically in Fig. 4c, showing all the cross section profiles and the continuous polar degree of freedom in the dowser state. The $D$ state itself has a remaining degree of freedom $\phi$, which forms a quasi-planar polar field with its own dynamics governed by the sine-Gordon equation[57] and may contain point defects and solitons, as explored in great detail in previous publications[33,35].

Qualitative analysis discussed above only outlines the topological nature of the orientational phases. To evaluate the phase transitions quantitatively, we performed director field simulations with a specific goal to investigate the effect of the flow amount and elastic anisotropy on the phase behaviour. As defects are not present in our system, and we favour a minimal model that can also yield analytical results, nematic elasticity is modelled with the Frank-Oseen elastic energy[22] $F = \int f \, dV$, where

$$f = \frac{K}{2}\left[(\nabla \cdot \mathbf{n})^2 + \kappa\left(\mathbf{n} \cdot (\nabla \times \mathbf{n}) - q_0\right)^2 + (\mathbf{n} \times (\nabla \times \mathbf{n}))^2\right]. \quad (1)$$

The ratio of twist to splay/bend elastic constants is measured by $\kappa = K_2/K$, assuming equality of splay and bend elastic constants $K_1 = K_3 = K$, and neglecting the saddle-splay contribution, which is constant in cases of strong non-degenerate surface anchoring. For 5CB in the nematic phase, the elastic constant ratio falls within the range $0.4 < \kappa < 0.6$. The flow rate is measured by the Ericksen number $\mathrm{Er}$. The intrinsic chirality $q_0$ is set to zero unless stated otherwise. The director field dynamics is simulated following the Ericksen-Leslie-Parodi equation[22] (see Methods).

One-dimensional (1D) simulations with respect to the vertical coordinate $z$ produce director profiles where the exact nature of the phase transition can be observed and mid-plane angle $\theta_0$, the order parameter of this transition, can be quantitatively measured. Figure 5 shows detailed dependence of the director angle with respect to the vertical position $z$. In the $B$ state, the $\theta$ dependence goes through 0 in the mid-plane and $\phi$ is a step function, and after the transition the mid-plane angle increases steeply. At high $\mathrm{Er}$ and small $\kappa$ (low twist elastic constant), the $\theta(z)$ and $\phi(z)$ dependences indicate a relatively uniform heliconical twist near the channel mid-plane.

Figure 6a shows the results of 1D simulations in the form of a phase diagram, revealing the region of stability of the $B^*$ phase with respect to the Ericksen number and the elastic constant anisotropy. The boundary between the phases fits well to an empirical power law in the form $\kappa = \kappa'[1 - (\mathrm{Er}/\mathrm{Er}')^{-\beta}]$ (white curve in Fig. 6a) with lowest admissible flow rate $\mathrm{Er}' = 9.87$ and critical elastic anisotropy $\kappa' = 0.97$ that is required for $B^*$ phase to occur, and the critical exponent $\beta = 1.4$. Note that a lower

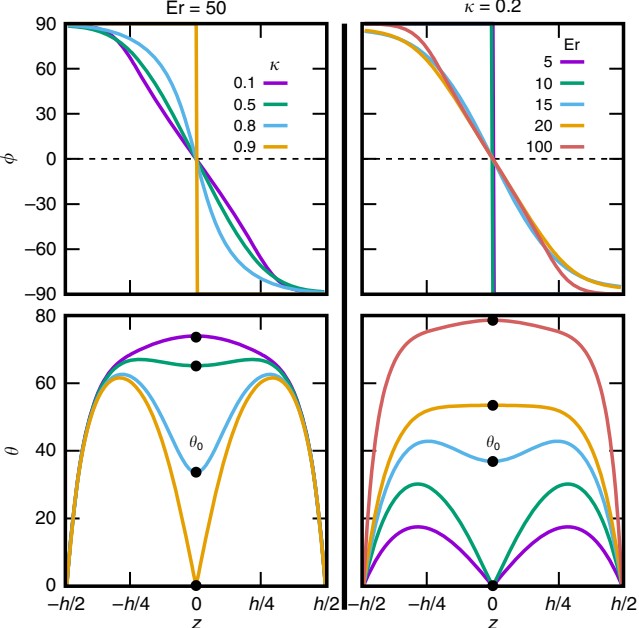

**Fig. 5 Vertical director profile variation with flow rate and elastic anisotropy.** Twist angle $\phi$ and tilt angle $\theta$, revealed by numerical integration of Eq. (8) (see Methods) for varying elastic anisotropy. $\kappa$ (left column) and Ericksen number Er (right column). At small elastic anisotropies, director structure is in the achiral $B$ state, characterised by a sharp transition from $\phi = 90°$ to $\phi = -90°$ and large director tilt in the flow direction in the upper and bottom half of the channel. As $\kappa$ is reduced, $\theta_0$ in the middle of the channel becomes non-zero and $\phi$ profile smooths out, indicating a chiral transition from the director tilt in the bottom and in the upper half, characteristic for the $B^*$ configuration, as already discussed in Fig. 4. Similarly, director is in the $B$ state for small values of Er. The director tilt angle increases with increasing Er, until a transition to $B^*$ state is reached.

twist elastic constant $\kappa < \kappa' < 1$ is required to stabilise the $B^*$ state, so simplified models with a single-elastic-constant approximation[54–56] cannot account for this phenomenon. A more elaborate Landau model, discussed in the next section, predicts the phase boundary consistent to the simulation result (red curve in Fig. 6a). Above the phase transition, the mid-plane angle $\theta_0$ increases steeply with an increasing Er and decreasing $\kappa$, with the critical exponent close to 0.5 (Fig. 6b, c). In previous work[35], we showed that $D$ state can be stabilised for Ericksen numbers higher than $\mathrm{Er}_D = \frac{\pi^3}{2(1+\lambda)} = 7.56$, which is lower than any Er that can accommodate the $B^*$ phase. Under this choice of parameters, the chiral $B^*$ phase is never a true steady state. This may explain why, to our best knowledge, there is no description of this state in the past literature. However, the transition to the $D$ phase is discontinuous and requires an initial seed domain of sufficient size for it to grow and engulf the entire sample[35]. Therefore, it is possible, with careful control of flow rate, to observe a transition to the chiral $B^*$ structure despite it being only metastable with respect to the $D$ state.

With the addition of a chiral dopant ($q_0 \neq 0$), the typical pitchfork bifurcation in $\theta_0$ at the second-order phase transition splits into two branches. Figure 6d shows the smooth dependence of $\theta_0$ on the flow rate for the state with favourable chirality and existence of metastable states with wrong chirality, which discontinuously transforms into its mirror image when the Ericksen number falls below a certain threshold. This is a typical behaviour of a second-order phase transition, linearly coupled to an external field. The $\theta_0(\mathrm{Er})$ behaves linearly for

small Er, with the slope proportional to $q_0$. Very small concentrations of the chiral dopant, with intrinsic chiral pitch thousands of times longer than the height of the channel, suffice to instigate a noticeable chiral response below the $B$ to $B^*$ phase transition.

In a finitely wide channel, the side walls affect the structure (Fig. 7a–d). The homeotropic anchoring at the walls favours the side-to-side mid-plane alignment, which is characteristic of the $B^*$ state. In the $B$ state, this is only a localised boundary-induced state and the bulk of the channel has predominantly vertical mid-plane director (Fig. 7b), so the distinction between the chiral and achiral phase is blurred in narrow channels with width comparable to the height. In the $B^\pm$ states, the walls ensure that the chiral state can stabilise across the entire channel width despite lower flow velocity at the side walls (Fig. 7c). Both chiral regions can coexist, with a narrow achiral soliton in the middle (Fig. 7d). Just like the mid-plane inclination, the width of the soliton that appears between the oppositely chiral states also undergoes singular behaviour at the phase transition. We simulated the soliton in a channel with aspect ratio 20 : 1 and observed variation of the vertical component of the director in the mid-plane: $n_z = \cos\theta\,(x, z = 0)$. The profile of the soliton fits well to the form dictated by the sine-Gordon equation, $n_z = \cos\theta_0 + (1 - \cos\theta_0)\cosh^{-1}(x/\xi)$, where $\theta_0$ is the previously mentioned mid-plane deviation in the uniform $B^*$ phase and $\xi$ is the characteristic width of the soliton (Fig. 7e). The width diverges with a critical exponent $b \approx 0.42$, as shown in Fig. 7f.

The director in channels subjected to flow thus undergoes a rich progression of states: a uniform vertical alignment is first 'bowed' in the direction of flow, then the mirror symmetry is broken in a second-order phase transition to produce a pair of discrete degenerate states, and finally a first-order phase transition changes the symmetry to a continuously varying in-plane angle. Whether the intermediate chiral state is observed before the escaped dowser state is reached depends on the twist elastic constant, steadiness of the flow, and the aspect ratio of the channel.

**Landau model of the $B$ to $B^*$ phase transition**. We constructed minimal Landau theory of a spatially uniform phase. Based on the simulation results and topological considerations, the simplest ansatz for both $B$ and $B^*$ phase is a director that traces an ellipse in projection to the $xy$ plane, parametrised by the vertical coordinate $z$:

$$\mathbf{n}(\tilde{z}) = \{a \sin 2\pi\tilde{z}, b(1 - \cos 2\pi\tilde{z}), n_z\}, \qquad (2)$$

where $\tilde{z} = \frac{1}{h}(z + \frac{h}{2})$ is the non-dimensional vertical position between 0 and 1; $a$ is the amplitude of the bowing into the flow direction, related to the maximal angle of flow-alignment in the $B$ phase: $a = \sin\theta_{\max}$, and $b$ represents the mid-plane tilt: $2b = \sin\theta_0$. The remaining vertical component $n_z$ of the director is uniquely determined by normalisation.

Up to the fourth order, the elastic free energy from Eq. (1), integrated over the thickness, reduces to

$$
\begin{aligned}
F_{\mathrm{el}} = \frac{K\pi^2}{2h}\bigg[ &2(a^2 + b^2) + \frac{1}{2}(a^4 + 5b^4) \\
&+ (6\kappa - 7)\,a^2 b^2 + \frac{8q_0 h}{\pi}\kappa a b \bigg] + \mathcal{O}(a^6, b^6)
\end{aligned}
\qquad (3)
$$

with sixth and higher order remainder terms omitted. In addition to elasticity, flow-alignment can be modelled phenomenologically with an additional term, proportional to Er and odd function of $a$, as flow reversal must reverse the direction of flow alignment[35]. We chose a cubic term $a - a^3/(3a_0^2)$ as the simplest odd function

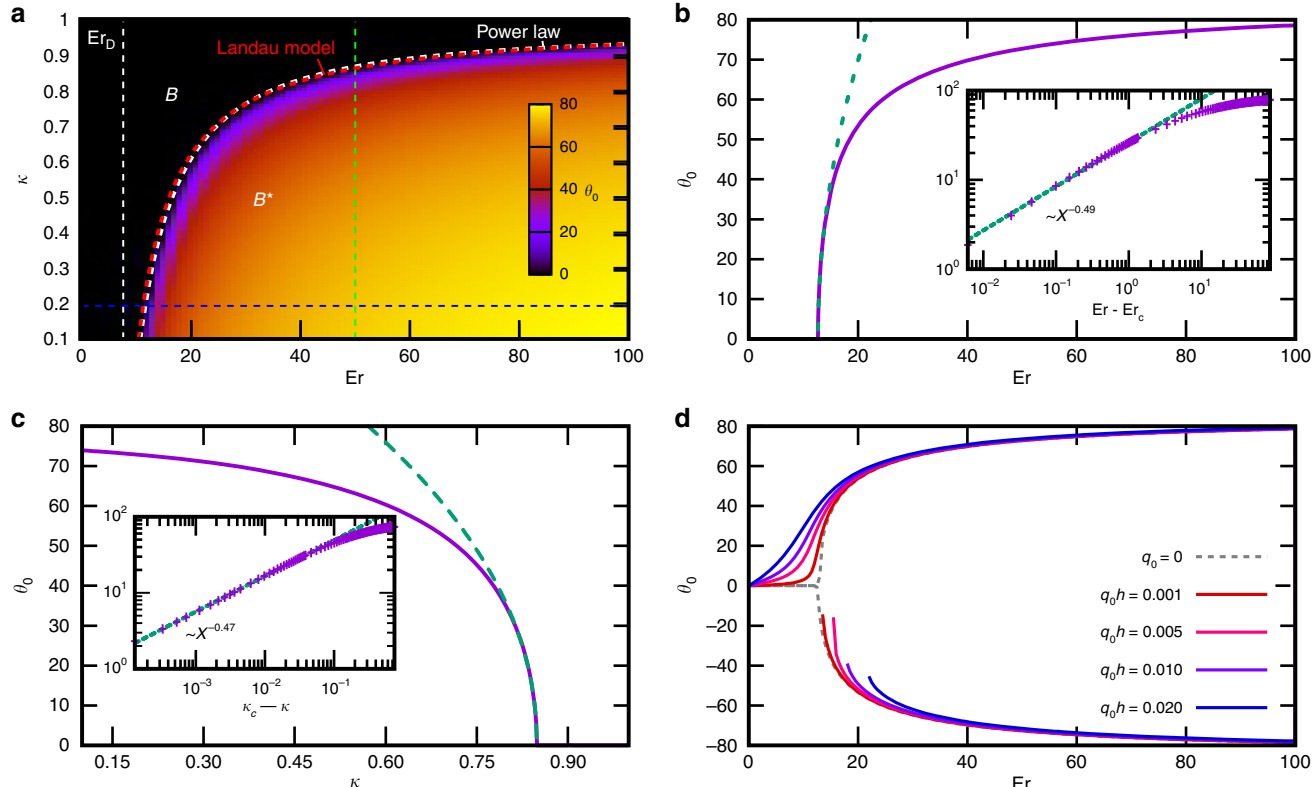

**Fig. 6 Phase transition to the chiral state B* with respect to elastic anisotropy κ and Ericksen number Er. a** $B$ state appears in the phase diagram at large values of $\kappa$ and at small values of Er. The $B^*$ state is characterised by non-zero mid-channel tilt angle $\theta_0$ of the director. The phase border is fitted with the function $\kappa = -A(\mathrm{Er} - C)^{-\beta} + D$ with exponent $\beta = 1.36$, asymptote at Er $= 9.87$ and $\kappa = 0.97$. Er$_D = 7.56$ indicates the value of Ericksen number at which $D$ phase can be stabilised. **b, c** Cross sections of the phase diagram at $\kappa = 0.2$ and Er $= 50$, respectively, show a clear continuous phase transition between $B$ and $B^*$ phase. In the vicinity of the transition region a power law reveals a critical exponent close to 0.5 (see insets). **d** Effect of intrinsic chirality on the mid-plane tilt angle $\theta_0$, obtained by numerical integration of Eq. (8). The symmetry of the pitchfork bifurcation at the second-order $B$ to $B^*$ phase transition is broken by a chiral dopant into stable and metastable branches. In a chiral system, the tilt angle has an immediate onset at small non-zero flow rates instead of an abrupt onset at the transition flow rate. The dotted line indicates the solution for no intrinsic pitch, shown also in **b**.

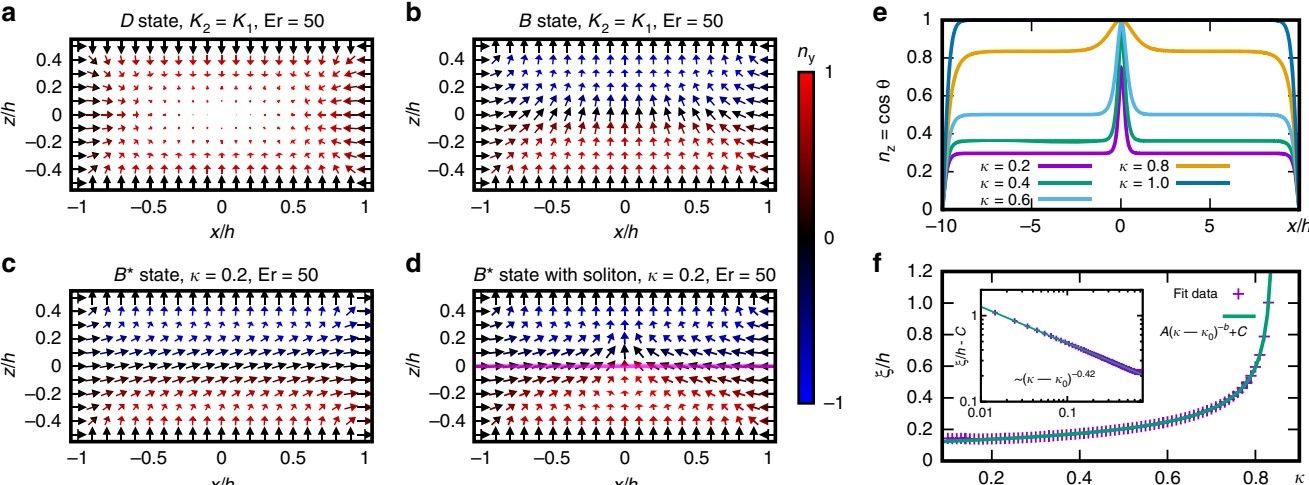

**Fig. 7 Effects of finite channel width on the stability of flow-induced patterns. a–d** 2D channel cross sections showing the director field of states $D$ in **a**, $B$ in **b**, and $B^*$ in **c, d**. The side walls force the director to point perpendicularly to the flow, inducing a chiral state near the wall. In $B$ state, this chirality transitions to the achiral profile away from the walls (**b**). In the $B^*$ state, it is continued from the walls across the channel with either a single chirality (**c**) or with opposite chiralities, meeting at the soliton in the middle (**d**). **e** Vertical component $n_z = \cos\theta$ of the director for different elastic anisotropy values $\kappa$ sampled along the mid-line (magenta line in **d**) shows the achiral soliton in the middle of the channel, where left- and right-handed $B^*$ domains meet. **f** Critical behaviour of the soliton width $\xi$ with respect to the elastic anisotropy $\kappa$. The inset shows the log-log plot of the power-law part of the expression, with the critical exponent $b \approx 0.42$.

with a single local minimum at $a_0 > 0$, which should coincide with the known steady state in the $\text{Er} \gg 1$ regime where elastic deformations introduced by the confinement can be neglected.

Determining the position of the minimum $a_0$ requires some care. Our sinusoidal model does not describe well the texture with a constant alignment at the Leslie angle $\vartheta_L = 8.9°$ with respect to the flow direction. As a reasonable approximation, we set the parameter to $a_0^2 = 2\cos^2\theta_L$, so that the average value of $|n_x|^2$ equals $|\cos\theta_L|^2$. We keep in mind that this extreme alignment will not occur when all terms are considered, and that with series expansion, we neglected the normalisation condition $|a| < 1$.

Overall, the phase behaviour is determined by the minima of the non-dimensional functional

$$\mathcal{F} = A\,\text{Er}\,\left[a\left[1 - \frac{a^2}{6\cos^2\vartheta_L}\right] + 2\left(a^2 + b^2\right)\right. \\ \left. + \frac{1}{2}\left(a^4 + 5b^4\right) + (6\kappa - 7)\,a^2b^2 + 8N\kappa ab,\right. \tag{4}$$

where $A$ is the only remaining free parameter which simply renormalizes the arbitrary numerical prefactor in the definition of the Ericksen number. This expression is a low order expansion to capture the onset of the phase transition. In order to actually predict the magnitude of $b$ in the $B^*$ phase, sixth order terms (predominantly the $b^6$ term in the splay elastic energy expansion) need to be introduced to ensure the free energy functional is bounded from below and the minimisation condition is well posed. With this addition, the critical growth of $b$ with the critical exponent 0.5 above the transition, as witnessed in Fig. 6b, c is recovered.

In the absence of a chiral dopant ($N = 0$), the phase boundary between $B$ and $B^*$ phase in terms of $\kappa$ and $\text{Er}$ can be retrieved without the higher order terms, by finding when the minimum of the free energy turns into a saddle point:

$$\left(\frac{\partial\mathcal{F}}{\partial a}\right)_{b=0} = 0, \quad \left(\frac{\partial^2\mathcal{F}}{\partial b^2}\right)_{b=0} = 0. \tag{5}$$

This system of equations resolves into the form

$$(A\text{Er})^2 = \frac{32(8 - 6\kappa)^2}{(7 - 6\kappa)(7 - \cos^{-2}\vartheta_L - 6\kappa)^2}. \tag{6}$$

Fitting the sole parameter $A = 0.297$ to the simulation data, shown in Fig. 6a, produces an excellent agreement with the simulation and the empirical power law expression; the fit is represented by the red dash-dotted curve.

In the limit of low flow rate, the bowing amplitude $a$ and mid-plane tilt $b$ respond approximately linearly,

$$a \approx \frac{A\,\text{Er}}{4}\left(1 - (2\kappa N)^2\right)^{-1}, \quad \frac{b}{a} \approx 2\kappa N, \tag{7}$$

where we took into account the chirality measure $N$, which acts through the linear coupling term $\sim Nab$. The bowing amplitude is minimally affected by chirality, but the mirror symmetry breaking triggers the chiral $B^*$ phase behaviour. The slope of $\theta_0$ with respect to $\text{Er}$ at slow flows grows linearly with the chiral pitch, as observed by simulations in Fig. 6d.

## Discussion

In this paper we demonstrated techniques of experimental control of a NLC flow in microconfinement to predictably produce desired orientational regimes and transitions between them. We focused particularly on the secondary orientational transition in the shadow of the main flow-aligning transition of nematic materials in homeotropic channels. This delicate state breaks the mirror symmetry, and exhibits rich dynamics of its left- and right-handed domains and the solitons between them. The main

parameter driving the transitions is flow velocity, which can be influenced by tuning the shape of the channel, or varying the driving pressure. We have provided a topological description of the director profiles before and after the transition and used numerical simulations to demonstrate the effect of the twist elastic constant and the crucial role it plays in stabilisation of the chiral state. The $B$ to $B^*$ transition behaves as a typical second-order phase transition and intrinsic chirality introduced through a dopant plays a role of a linearly coupled external field. The flow-aligned dowser field has been shown to exhibit interesting interactions with symmetry-breaking external effects, such as thickness gradient, channel branching and electric field[32–34,57], anchoring pretilt[53], weaker surface anchoring, and hybrid boundary conditions[54], opening a question for future investigations on how the pretransitional $B$ and $B^*$ states respond to the same stimuli, and how to widen the stability range of the $B^*$ state.

Hidden metastable phases and phases with narrow range of stability can prove to be interesting both theoretically and technologically, the blue phases being one example. In lyotropic and active nematics, additional behaviour is expected due to density- and activity-dependence and generally different elastic properties. Chiral structures in this paper could perform as bistable optical filters, where switching between left- and right-handed structure is induced by flow or by laser pulses. If the system is intrinsically chiral, the energy degeneracy will break, resulting in preference in one of the phases, and forces acting on the soliton. Soliton dynamics could be exploited in sensors to detect chiral molecules in real time during flow, which is a relevant ability in living biological mixtures. Further, we show how observed structures change the hydraulic conductivity in microchannels, which could be used in flow steering applications. Created solitons separating domains of opposite chirality could be utilized to trap and release small colloidal particles, much like disclination lines are known to do[27]. Finally, our work explores how fluids with nematic order form orientational structures in shear flow, where precise control of otherwise possibly highly irregular flow conditions is utilised by a microfluidic confinement.

## Methods

**Materials and experimental procedures**. The experimental protocol employed in the fabrication of microfluidic channels is based on the setup described in previous publications[31,35]. Our experiments begin with preparation of microfluidic channels with a rectangular cross section with height $h \approx 12\,\mu\text{m}$, width $w \approx 100\,\mu\text{m}$ and length $L \approx 20\,\text{mm}$. The channels were fabricated out of polydimethylsiloxane (PDMS, Sylgard 184, Dow Corning) reliefs and pristine cover glasses (Hirschmann Laborgeräte) by following standard soft lithography procedures[21]. For experiments with laser tweezers, the cover glass was replaced by indium-thin-oxide (ITO) coated glass substrate (Xinyan Technology) which efficiently absorbs the infrared (IR) laser light, and provides precise control over the local heating of applied liquids. The lithographic design was not limited to linear channel geometry as we employed variable channel width with expansion chambers for a particular set of experiments. The channel walls were chemically treated with 0.2 wt% aqueous solution of N-dimethyl-n-octadecyl-3-aminopropyl-trimethoxysilyl chloride (DMOAP, ABCR), and finally dried with blowing hot air to render PDMS surface and glass substrate hydrophobic. This surface coating is known to induce strong perpendicular (homeotropic) alignment of a single-component NLC 4-cyano-4′-pentyl-1,1′-biphenyl (5CB, Synthon Chemicals) at the interfaces[31]. The functionalized channels were filled up with 5CB in isotropic phase (above 35 °C), which then gradually cooled down to nematic phase at room temperature. For some experiments, weakly chiral nematic medium was prepared by mixing 5CB host with 0.08 wt% right-handed chiral dopant 4-(2-methylbutyl)-4′-cyanobiphenyl (CB15, Merck) in isotropic phase. The value of helical pitch was set to 170 μm at the chosen concentration of CB15, which gives $\frac{7}{6}\pi$ rotation of 5CB molecules across the channel width. The concentration of the chiral dopant was too low to significantly alter the elastic constants or to induce a noticeable change in the optical texture in the absence of flow, but it was high enough to break the symmetry due to preference over one sense of twist, which resulted in altered response to pressure-driven flows. All the experiments were conducted at room temperature.

**Flow characterisation**. We drove and precisely controlled the nematic flows in microchannels by using piezoelectric pressure control system (OB1 MK3,

Elveflow). The flow rates were varied in the range 0.05 to 0.85 μL h$^{-1}$ corresponding to a flow velocity $v$, ranging from 5 to 150 μm s$^{-1}$. We typically used inbuilt flow profile routines for the controller that gradually increased and subsequently maintained the applied driving pressure. Poiseuille flow of a nematic fluid inside rectangular channels was strictly laminar with estimated Reynolds number Re $= \rho v h / \eta$ between $10^{-6}$ and $10^{-4}$ (taking density $\rho \approx 1.024$ kg m$^{-3}$ and effective dynamic viscosity[34] $\eta \approx 50$ mPa s). The Ericksen number Er $= \gamma v h / K$ varied roughly between 0.7 and 30 (taking $K \approx 5.5$ pN in the single-elastic-constant approximation and rotational viscosity[22] $\gamma \approx 81$ mPa s). Note that the definition of the Reynolds and Ericksen number may vary by a constant depending on the choice of typical reference values. For example, we took the channel thickness $h$ as the typical length scale, as the width of the channels varies between experiments.

**Optical microscopy and laser tweezers manipulation.** The reorientational dynamics of the 5CB under microfluidic confinement was studied by an inverted polarised light microscope (Eclipse Ti-U, Nikon), equipped with CFI Plan ×2, ×10 and ×20 objectives. The samples were observed between crossed polarizers in a transmission mode. The channels were typically oriented at 45° with respect to the polarizers to obtain wider selection of transmitted interference colours. In addition, we used a laser tweezers setup build around the inverted optical microscope with IR fiber laser operating at 1064 nm as linearly polarised light source. A pair of acousto-optic deflectors, driven by computerised system (Tweez200si, Aresis), was used for precise laser beam manipulation. The laser power of a diffraction limited Gaussian beam in the sample plane was tuned from 20 to 200 mW, and it was primarily used for local heating of the 5CB above its clearing temperature. The nematodynamics was recorded in full HD colour videos using digital CMOS camera (EOS 750D, Canon) at a frame rate of 30 frames per second. The image analysis was performed by using multi-purpose video tools Fiji and VideoMach.

**Numerical simulations.** We performed numerical simulations using a director field model. Such approach is valid as the studied textures do not contain defects or significant variations in the degree of nematic order, and the director-based approach offers speed and simplicity over Q-tensor models. The director field dynamics is given by the Ericksen-Leslie-Parodi equation[22]

$$\dot{\mathbf{n}} + \omega \mathbf{n} = \Pi \left( \lambda u \mathbf{n} + \frac{1}{\gamma} \mathbf{h} \right), \qquad (8)$$

where $\omega_{ij} = \frac{1}{2} (\partial_i v_j - \partial_j v_i)$ and $u_{ij} = \frac{1}{2} (\partial_i v_j + \partial_j v_i)$ are the vorticity and the strain tensor, respectively. $\Pi_{ij} = \delta_{ij} - n_i n_j$ is the projection operator, $\mathbf{h} = -\frac{\delta F}{\delta \mathbf{n}}$ is the molecular field, $\lambda$ the alignment parameter, and $\gamma$ the rotational viscosity. For a typical velocity $v$, length scale equal to the height of the channel $h$ and fixed boundary conditions, Eq. (8) is governed by only four parameters: (i) $\lambda$, which determines the alignment response of nematic molecules in shear flows, including the preferred tilt angle with respect to the flow direction, often called Leslie angle $\vartheta_L$, (ii) Ericksen number Er $= \gamma v h / K$, which compares viscous to elastic forces, (iii) elastic anisotropy $\kappa = K_2 / K$ and (iv) the chirality measured by the number of pitch lengths per thickness $N = \pi^{-1} q_0 h$.

In our simulations and analytical calculations we assume a steady-state Poiseuille flow profile through the channel—in line with the large aspect ratio of the channels in experiments ($w/h \approx 8$)—and neglect the influence of the orientational order on the flow profile (i.e. backflow). We fix $\lambda = 1.05$ and $\vartheta_L = 8.9°$ which corresponds to viscosity parameters of 5CB[22], and calculate stationary director profiles for different values of the flow rate (expressed through Er and $\kappa$) by numerically integrating Eq. (8) until a steady state is reached. We simulate 2D channel cross sections to model solitons and channel edge effects, and 1D vertical profiles for phase behaviour in homogeneous domains, both with a finite difference method.

## Data availability
The data that support the findings of this study are available from the corresponding author upon reasonable request.

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

## Acknowledgements

We thank Miha Ravnik for beneficial discussions and support. We acknowledge funding by the Slovenian Research Agency (ARRS) under contracts P1-0099 (to S.Č. and Ž.K.), P1-0055 (to T.E. and U.T.), L1-8135 (to S.Č., Ž.K., and U.T.), J1-9149 (to S.Č and Ž.K.), and N1-0124 (to Ž.K.). S.Č. and U.T. thank COST Action CA17139 for supporting their research activities.

## Author contributions

U.T. and S.Č. designed the research. T.E. and U.T. conducted the experiments and analysed data. S.Č. and Ž.K. performed the numerical simulations, developed theoretical model, and analysed the results. U.T. coordinated the research and supervised the experiments. S.Č., Ž.K., and U.T. wrote the paper. All authors discussed the progress of research and contributed to the final version of the paper.

## Competing interests

The authors declare no competing interests.
