## [Peer Review File · Nature Communications]

Reviewers' comments:

Reviewer #1 (Remarks to the Author):

This paper reports the results of a detailed study of intermediate chiral orientational structures that occurs in a microfluidic flow of nematic liquid crystals. Employing a precision pressure regulator, the authors were able to experiment with the spontaneously occurring chiral states and explained the observation using a simple Landau type analysis and numerical simulations. The results show an unexplored intricacy of coupling between flow and orientation in liquid crystal hydrodynamics. I think the paper deserves publication, provided the points described below are considered and are properly taken into account.

1. It is certainly a technical advance to use a precise pressure regulator to be able to access metastable states confined in a narrow operation range. With that said, it is one of the most intriguing and salient points in microfluidics of complex fluids to explore the pressure-flow rate relationship with a high precision. Hydrodynamic nonlinearity associated with structural evolution is of fundamental significance in understanding the flow behaviors. A relevant plot should be included and the results have to be compared with the theoretical estimates.

2. Although the transition to D state is not the primary target of this paper, it seems appropriate to give more experimental details about the transition. Since the transition involves a generation of a disclination line and the disclination line introduces additional viscous dissipation, the knowledge about the transition gives a background indicating the subtle nature of the chiral intermediate states under discussion. The pressure-flow rate relationship is also relevant for this transition as well.

Reviewer #2 (Remarks to the Author):

Review of Copar et al.

This is an experimental paper about how flow affects the alignment of nematic liquid crystals. In a microfluidic channel in which the walls have been treated to make the rod-like molecule align perpendicular to the walls, there is a discontinuous transition to a flow aligned state at sufficient pressure. If the twist elastic constant is sufficiently low, there is a chiral intermediate state that arises before the transition. This state is the subject of the manuscript.

The topic is timely, since the interplay of flow and nematic alignment is currently of great interest for studies of both passive and active matter. As the authors emphasize, other groups have recently studied chiral states that form from achiral molecules in equilibrium systems; this is the first example I have seen of a chiral state forming from achiral molecules in a flow situation. The paper is thorough and complete. The authors not only use flow to induce the intermediate state, but also study the symmetry-breaking effect of chiral dopants, the dependence of the transition on channel geometry, and the effect of laser tweezer perturbations. The experiments are supported by theory and some simulations. I found the paper to be well-written. I recommend publication. I only have minor comments.

I realize it is a losing battle, but "achiral symmetry breaking" makes more sense than "chiral symmetry breaking". Or better still: "mirror symmetry breaking."

I found figures 1f--1h slightly confusing. Are each of f, g, and h instances of the emergence of the

B*/B+/B- states for different sequence of increasing the flow velocity, all at the acceleration of $15 \text{ } \mu\text{m/s}^2$? What does the yellow dotted line mean in the 3rd figure of 1f and the upper two panels of 1h?

Figures 1 and 4 would be clearer with actual axes x and z instead of the notation xz to denote the plane being viewed. The reader can eventually figure out what z is by context but there is enough complicated geometry in this paper already, so why not make the easy things easy. Also, I had to stare at figure 4a for a minute before I thought I understood it. Why not help the reader by stating that B+ and B- are mirror images of each other, with the mirror in the xz plane?

Reviewer #3 (Remarks to the Author):

The authors report the possible topological states in channel confined nematic flows that arise in real time during flow. They report that in microfluidic channels with perpendicular surface alignment, nematics discontinuously transition from perpendicular structure at low flow rates to flow aligned structure at high flow rates. They have also carried out some experiments in order to support their simulated results on topologically protected chiral intermediate state and have constructed a phenomenological model to explain the B to B* phase transition. Though the authors have carried out both experimental and simulation work to explain that B state undergoes a continuous symmetry breaking transformation into B* state they have ignored the effect of surface anchoring term. The manuscript needs a revision before it can be accepted for publication. The points for revision are as follows:

Authors have carried out the simulations based on Frank-Oseen elastic energy model with one constant approximation ignoring K₂₄ term. Also they have not taken into consideration the surface free energy due to anchoring into account while investigating the B to B* phase transition. In the experimental methods the authors just mention that concentration of chiral dopant was too low.....(line 526 to 529 of pg 8 of the manuscript). The authors can provide the value of pitch and the change in twist elastic constant on doping with chiral dopant as these factors also play a main role on the experimental results.

Reviewer #4 (Remarks to the Author):

The authors have carried out some very nice experimental work (the supplementary movies are particularly nice), and have provided a thorough characterization of the multiple flow and orientation regimes that arise in their experimental u-fluidic system with NLC. They also propose a simple (partly phenomenological) energetic model, predictions of which are compared to their experiments.

The work is interesting, novel, and worthy of publication, but I do have several comments. Though the Introduction talks in general terms about the increasing miniaturization of devices, and using u-fluidics for control, it is not really elucidated how the results of the paper, even hypothetically, might be harnessed. The last 7 line of the Discussion mention a possible application, but for publication in a high-profile journal such as Nature Comms I would like to see evidence of more widespread applicability of the results, which would indicate relevance to a wide audience.

The authors could also improve on the explanation of the results presented. For figures where director orientation is shown, it should be clearly stated in the caption or text whether the orientation is a

sketch, or obtained from simulation, and if so, reference to the model solved should be provided. For example, it's not clear to me whether Figure 4 was obtained by solving the model in the Methods section, or if it is understood to be schematic. The figure caption actually suggests the latter, but it's unclear.

A few additional specific comments (some only minor) follow below:

- Line 95: "horizontally" - might be more meaningful to say the director is parallel to the channel walls since the reader has no idea how the channel is oriented in the lab.

- Line 326: "empirical power law" - how is this illuminating? As presented, it sounds just like curve-fitting. Is there any insight that the reader is supposed to take from this model? How does it help understanding of the system?

- Lines 319, 337: Why are results discussed in terms of K_2 and not κ ?

- Eq(2) and subsequently: Is b here the same as in the power law of line 326? Assuming not, it is confusing to have the same notation.

- Lines 413, 414: "with an additional term..." - please be more specific. Do you mean that eq.(3) should be supplemented by an additional term? Giving eq.(4)? It really should not be as difficult to follow as it is, if so. If the choice of a cubic term is phenomenological, can the authors provide any further justification?

Microfluidic control over topological states in channel-confined nematic flows

submitted by:

Simon Čopar, Žiga Kos, Tadej Emeršič, Uroš Tkalec

Response to Reviews

Reviewer 1

Comments to the Authors: *This paper reports the results of a detailed study of intermediate chiral orientational structures that occur in a microfluidic flow of nematic liquid crystals. Employing a precision pressure regulator, the authors were able to experiment with the spontaneously occurring chiral states and explained the observation using a simple Landau type analysis and numerical simulations. The results show an unexplored intricacy of coupling between flow and orientation in liquid crystal hydrodynamics. I think the paper deserves publication, provided the points described below are considered and are properly taken into account.*

Authors' answer: We thank the reviewer for the appreciation of the manuscript and positive opinion of our results.

Reviewer's comment # 1: *It is certainly a technical advance to use a precise pressure regulator to be able to access metastable states confined in a narrow operation range. With that said, it is one of the most intriguing and salient points in microfluidics of complex fluids to explore the pressure-flow rate relationship with a high precision. Hydrodynamic nonlinearity associated with structural evolution is of fundamental significance in understanding the flow behaviors. A relevant plot should be included and the results have to be compared with the theoretical estimates.*

Authors' answer: We thank the reviewer for his or her comments and the relevant suggestion. The flow of nematic 5CB, induced by the application of a pressure gradient in microfluidic channels was extensively studied by Sengupta and coworkers (see Refs. [21, 27, 31, 34]). In Ref. [21] (Page 24), the authors presented measured mass flow rate as a function of applied pressure difference in homeotropic channels, which was obtained by a precise gear pump. They discussed how the boundary conditions on the channel walls influence the effective viscosity of a flowing nematic, but

did not relate the response to different flow states. More detailed 2D lattice Boltzmann simulation results on driven nematic flows in homeotropic confinement have been presented by Batista *et al.*, in Ref. [54], where they found a discontinuity in flow rate for strong homeotropic anchoring.

We have now performed similar experimental verification of the flow velocity-pressure relationship in view of underlying structural transitions. We observed a discontinuity in flow velocity (and thus effective viscosity) at the B to B^* transition, but no considerable change during B^* to D transition. These additional results were added into the Electronic Supplementary Material, together with interpretation of the results. The Supplementary Fig. 1 is also included below for convenience.

Supplementary Figure 1: **Flow velocity as a function of the stationary pressure difference.** For caption and discussion, see Electronic Supplementary Material.

Reviewer’s comment # 2: *Although the transition to D state is not the primary target of this paper, it seems appropriate to give more experimental details about the transition. Since the transition involves a generation of a disclination line and the disclination line introduces additional viscous dissipation, the knowledge about the transition gives a background indicating the subtle nature of the chiral intermediate states under discussion. The pressure-flow rate relationship is also relevant for this transition as well.*

Authors’ answer: We thank the reviewer for the suggestion. As discussed in the response to the previous comment, we performed additional experiments to obtain the pressure-flow velocity characteristics. As we are estimating flow velocity by tracking objects, we do not have time resolution to detect the passage of the disclination line, only steady-state regime before and after the transition.

Additional results on time dependence of the state in the middle of the channel during different flow acceleration rates, from which we can draw conclusions about lifetime and stability of the states under different conditions, are given by Supplementary Fig. 2.

Reviewer 2

Comments to the Authors: *This is an experimental paper about how flow affects the alignment of nematic liquid crystals. In a microfluidic channel in which the walls have been treated to make the rod-like molecules align perpendicular to the walls, there is a discontinuous transition to a flow aligned state at sufficient pressure. If the twist elastic constant is sufficiently low, there is a chiral intermediate state that arises before the transition. This state is the subject of the manuscript.*

The topic is timely, since the interplay of flow and nematic alignment is currently of great interest for studies of both passive and active matter. As the authors emphasize, other groups have recently studied chiral states that form

from achiral molecules in equilibrium systems; this is the first example I have seen of a chiral state forming from achiral molecules in a flow situation. The paper is thorough and complete. The authors not only use flow to induce the intermediate state, but also study the symmetry-breaking effect of chiral dopants, the dependence of the transition on channel geometry, and the effect of laser tweezers perturbations. The experiments are supported by theory and some simulations. I found the paper to be well-written. I recommend publication. I only have minor comments.

Authors' answer: We acknowledge the encouraging comments of the reviewer. We have addressed questions and comments in the revised manuscript.

Reviewer's comment # 1: *I realize it is a losing battle, but “achiral symmetry breaking” makes more sense than “chiral symmetry breaking”. Or better still: “mirror symmetry breaking”.*

Authors' answer: We thank the referee for this remark. We revised the expression by following the suggestion.

Reviewer's comment # 2: *I found figures 1f–h slightly confusing. Are each of f, g, and h instances of the emergence of the $B^*/B^+/B^-$ states for different sequence of increasing the flow velocity, all at the acceleration of $15 \mu\text{m}/\text{s}^2$? What does the yellow dotted line mean in the 3rd figure of 1f and the upper two panels of 1h?*

Authors' answer: We thank the reviewer for this precise observation and the suggested clarification. We performed additional experiments on the emergence of different flow states at various pressure differences and flow accelerations. The results are summarized in Supplementary Fig. 2. The emergence of chiral states in Fig. 1, f-h, happened at approximately same acceleration rate though the time to reach terminal flow velocity differed. We removed the yellow dotted lines as it is evident how fronts of the chiral phase propagate.

Reviewer's comment # 3: *Figures 1 and 4 would be clearer with actual axes x and z instead of the notation xz to denote the plane being viewed. The reader can eventually figure out what z is by context but there is enough complicated geometry in this paper already, so why not make the easy things easy. Also, I had to stare at figure 4a for a minute before I thought I understood it. Why not help the reader by stating that B^+ and B^- are mirror images of each other, with the mirror in the xz plane?*

Authors' answer: We appreciate the suggestion for clarification. We added xyz axes to Fig. 1a and Fig. 4a. We improved the description of symmetry relation between B^+ and B^- states in the main text. The difference was already notified in a caption to Fig. 1.

Reviewer 3

Comments to the Authors: *The authors report the possible topological states in channel confined nematic flows that arise in real time during flow. They report that in microfluidic channels with perpendicular surface alignment, nematics discontinuously transition from perpendicular structure at low flow rates to flow aligned structure at high flow rates. They have also carried out some experiments in order to support their simulated results on topologically protected chiral intermediate state and have constructed a phenomenological model to explain the B to B^* phase transition. Though the authors have carried out both experimental and simulation work to explain that B state undergoes a continuous symmetry breaking transformation into B^* state they have ignored the effect of surface anchoring term. The manuscript needs a revision before it can be accepted for publication. The points for revision are as follows:*

Reviewer's comment # 1: *Authors have carried out the simulations based on Frank-Oseen elastic energy model with one constant approximation ignoring K_{24} term. Also they have not taken into consideration the surface free energy due to anchoring into account while investigating the B to B^* phase transition.*

Authors' answer: We thank the reviewer for the comments on the paper. In our simulations and theoretical analysis, we were looking for the *simplest* model that explains the emergence of the symmetry-breaking flow regime and its qualitative behaviour. Therefore, we used Frank-Oseen elastic energy with two different elastic constants; K_2 is varied while K_1 and K_3 remain equal to each other.

As noted by the reviewer, K_{24} and surface free energy were indeed left out from the simulations and theoretical analysis. Besides our motivation to use the simplest model possible to explain the experimental observations, saddle-splay and surface free energy can be omitted, because surface anchoring in our experiments is indeed very strong; see Refs. [31, 35], and A. Sengupta *et al.*, *Microfluid Nanofluid* **13**, 941 (2012). K_{24} contribution is only significant, if the director has enough orientational freedom at the interface to change its saddle-splay deformation; for instance, in planar degenerate anchoring; see Ž. Kos and M. Ravnik, *Soft Matter* **12**, 1313 (2016). Strong homeotropic surface anchoring in our experiments prevents the saddle-splay elasticity from affecting the structure.

If homeotropic surface anchoring would be indeed weaker, we could expect: (i) higher flow rates at comparable pressure difference (see Ref. [54]), (ii) shifted lines in phase diagram (for example, Ref. [54] states that with weaker surface anchoring, the transition to the D state would be less favourable), or (iii) considerable K_{24} contribution, which is beyond the scope of this paper. However, saddle-splay contribution is now addressed in the main text (Page 5, Lines 315-318), and weaker surface anchoring is now mentioned in Discussion section (Page 8, Lines 495-498).

Reviewer's comment # 2: *In the experimental methods the authors just mention that concentration of chiral dopant was too low ... (line 526 to 529 of pg 8 of the manuscript). The authors can provide the value of pitch and the change in twist elastic constant on doping with chiral dopant as these factors also play a main role on the experimental results.*

Authors' answer: We thank the reviewer for this suggestion. We updated the Methods section to provide the pitch value, which is 1.7 times the width of the channel. The helical twisting power of the chiral liquid crystal mixture was determined by $p = (\xi c)^{-1}$, where ξ is the helical twisting power of the chiral dopant CB15 ($\xi \approx 7.3 \mu\text{m}^{-1}$ in 5CB), and c is the mass fraction of the dopant (0.08 wt% in our case). The ξ value of CB15 in 5CB was obtained from Trivedi *et al.*, *Proc. Nat. Acad. Sci. USA* **109**, 4744 (2012). Experimental measurements of the pitch length were, for instance, reported by E. P. Raynes, *Liq. Cryst.* **33**, 1215 (2006).

Regarding the effect of the chiral dopant, we realize the statement in the manuscript was misleading – the symmetry breaking is not due to change of K_1 , K_2 , K_3 but instead by changing the helical twisting power. We made changes to rectify this mistake (Page 8, Lines 550-554).

The addition of dopant will in general reduce all elastic constants. Example for E8 + CB15 was measured by Fedak *et al.*, *Mol. Cryst. Liq. Cryst.* **82**, 173 (1982), which was followed by other researchers, such as Watson *et al.*, *Liq. Cryst.* **28**, 1 (2001), to estimate the effect of the dopant in other mixtures.

Due to very low concentrations, we expect the effect to be smaller than that of temperature variations or other approximations taken in our model (such as $K_1 \approx K_3$), and thus safely neglected. The ratio of K_2 to other elastic constants (κ), which is the more important parameter in our case, should be even less affected than the constants themselves, especially at concentrations that we are dealing with.

Reviewer 4

Comments to the Authors: *The authors have carried out some very nice experimental work (the supplementary movies are particularly nice), and have provided a thorough characterization of the multiple flow and orientation regimes that arise in their experimental μ -fluidic system with NLC. They also propose a simple (partly phenomenological) energetic model, predictions of which are compared to their experiments.*

Reviewer's comment # 1: *The work is interesting, novel, and worthy of publication, but I do have several comments. Though the Introduction talks in general terms about the increasing miniaturization of devices, and using μ -fluidics*

for control, it is not really elucidated how the results of the paper, even hypothetically, might be harnessed. The last 7 line of the Discussion mention a possible application, but for publication in a high-profile journal such as Nature Comms I would like to see evidence of more widespread applicability of the results, which would indicate relevance to a wide audience.

Authors' answer: We thank the reviewer for the comment. We now address more potential applications of our results in the last paragraph of Discussion. Besides the relevance to lyotropic and active systems, four main potential applications are discussed: (i) bistable optical filters, where switching between left- or right-handed structure is driven by flow or by laser pulses; (ii) nematic microfluidic sensors for material chirality; (iii) flow steering by changing the metastable structure; (iv) trapping and releasing of small colloidal particles by solitons.

Reviewer's comment # 2: *The authors could also improve on the explanation of the results presented. For figures where director orientation is shown, it should be clearly stated in the caption or text whether the orientation is a sketch, or obtained from simulation, and if so, reference to the model solved should be provided. For example, its not clear to me whether Figure 4 was obtained by solving the model in the Methods section, or if it is understood to be schematic. The figure caption actually suggests the latter, but its unclear.*

Authors' answer: We thank for the comment. We now make it clear in the caption in Fig. 4 that it is a sketch that discusses the qualitative characteristics of all the phases.

Reviewer's comment # 3: *A few additional specific comments (some only minor) follow below:*

- Line 95: “horizontally” – might be more meaningful to say the director is parallel to the channel walls since the reader has no idea how the channel is oriented in the lab.
- Line 326: “empirical power law” – how is this illuminating? As presented, it sounds just like curve-fitting. Is there any insight that the reader is supposed to take from this model? How does it help understanding of the system?
- Lines 319, 337: Why are results discussed in terms of K_2 and not κ ?
- Eq (2) and subsequently: Is b here the same as in the power law of line 326? Assuming not, it is confusing to have the same notation.
- Lines 413, 414: “with an additional term . . .” – please be more specific. Do you mean that eq. (3) should be supplemented by an additional term? Giving eq. (4)? It really should not be as difficult to follow as it is, if so. If the choice of a cubic term is phenomenological, can the authors provide any further justification?

Authors' answer: We thank the reviewer for the specific comments. We address it as follows:

- Line 95: We replaced “horizontally” with “in the flow direction” to make it independent on the laboratory geometry.
- Line 326: It is indeed just curve-fitting, which gives us a way to predict the phase transition quantitatively, but does not illuminate the mechanism lying behind it (which is why we call it an empirical power law). We do, however, get the limiting values κ_0 and Er_0 , which is discussed in the same paragraph. A theory-backed model is developed later in the manuscript.
- Lines 319, 337: We replaced the remaining instances of K_2 with discussions about κ .
- Thank you for pointing that out, we changed the power law exponent to β to prevent confusion.
- Lines 413, 414: We now include the explicit equation for the extra cubic term, together with more elaborate justification on how we chose this particular form.

REVIEWERS' COMMENTS:

Reviewer #1 (Remarks to the Author):

With the additional experiments on the flow rate versus the pressure relation through the series of the transitions, my original concerns have been satisfactorily addressed. I am of the opinion that the paper is acceptable for publication.

Reviewer #2 (Remarks to the Author):

I am satisfied with the authors' revisions and responses to the referees.

Reviewer #3 (Remarks to the Author):

The authors have addressed the comments raised by the reviewers and have incorporated the relevant data in the revised manuscript or as supplementary data. I am in favor of publication of this manuscript in the present form.

Reviewer #4 (Remarks to the Author):

I am happy with the revisions the authors have made and approve acceptance.

Microfluidic control over topological states in channel-confined nematic flows

submitted and revised by:

Simon Čopar, Žiga Kos, Tadej Emeršič, Uroš Tkalec

Final Response to Reviews

Reviewer 1

Remarks to the Authors: *With the additional experiments on the flow rate versus the pressure relation through the series of the transitions, my original concerns have been satisfactorily addressed. I am of the opinion that the paper is acceptable for publication.*

Reviewer 2

Remarks to the Authors: *I am satisfied with the authors' revisions and responses to the referees.*

Reviewer 3

Remarks to the Authors: *The authors have addressed the comments raised by the reviewers and have incorporated the relevant data in the revised manuscript or as supplementary data. I am in favour of publication of this manuscript in the present form.*

Reviewer 4

Remarks to the Authors: *I am happy with the revisions the authors have made and approve acceptance.*